# Deep Eutectic Solvent-Based Ultrahigh Pressure Extraction of Baicalin from *Scutellaria baicalensis* Georgi

**DOI:** 10.3390/molecules23123233

**Published:** 2018-12-07

**Authors:** Hui Wang, Xiaodi Ma, Qibin Cheng, Li Wang, Liwei Zhang

**Affiliations:** 1Institute of Molecule Science, Key Laboratory of Chemical Biology and Molecular Engineering of Ministry of Education, Shanxi University, Taiyuan 030006, China; gxx125@126.com (H.W.); 15735170019@163.com (X.M.); qbcheng1992@163.com (Q.C.); 2Taiyuan University of Science and Technology, Taiyuan 030024, China; 3Institute of Environmental Science, Shanxi University, Taiyuan 030006, China; wangli@sxu.edu.cn

**Keywords:** baicalin, *Scutellaria baicalensis* Georgi, deep eutectic solvents, ultrahigh pressure extraction

## Abstract

Deep eutectic solvents (DESs), promising green solvents, and ultrahigh pressure extraction (UPE) as an effective auxiliary extraction method, have attracted wide attention. In this study, DES was coupled with UPE to efficiently extract baicalin from *Scutellaria baicalensis* Georgi. First, choline chloride: lactic acid (ChCl-LA, molar ratio 1:1) was selected as the most appropriate DES by comparing the extraction yield of different DESs. Second, the extraction protocol was optimized by response surface methodology (RSM) considering the impacts of ChCl-LA concentration, extraction pressure, extraction time and liquid-solid ratio on the extraction yield. Under the optimal condition (40 vol% water content, extraction pressure of 400 MPa, extraction time of 4 min and a liquid-solid ratio of 110 mL/g), a maximum yield of 116.8 mg/g was achieved, higher than that obtained by the traditional extraction method. The microstructure of the raw and extracted *Scutellaria baicalensis* Georgi samples according to scanning electron microscope (SEM) images revealed that the dissolution of chemical components was enhanced from the disrupted root tissues after DESs-UPE. DESs coupled with UPE could effectively extract the baicalin from *Scutellaria baicalensis* Georgi as a rapid and efficient extraction method.

## 1. Introduction

Ultrahigh pressure extraction (UPE) has many advantages, such as shorter processing times, higher extraction yields, lower power consumption, and less impurities in the extraction liquid [1]. As such UPE has been widely used in the casting industry, pharmaceutics, metallurgy, plastic making, civil engineering, and food industry. In addition, UPE can be conducted at low temperatures, and no chemical degradation reactions occur. This process could thus effectively avoid structural changes, losses, and the reduction of biological activity for the active ingredients [2]. Recently, UPE has been used for the extraction of catechins from green tea [3], phenolic compounds from longan fruit pericarp [4], lycopene from tomato paste waste [5], anthocyanins from grape skins [6], flavones and salidroside from *Rhodiola sachalinensis* [7], β-carotene from *Dunaliella salina* [8], 2-α-hydroxyursolic acid from crape myrtle leaf [9].

Deep eutectic solvents (DESs), have also attracted significant attention as promising green media. DESs are easily available by mixing a hydrogen bonding acceptor (HBA) with a hydrogen bonding donor (HBD) and after continuous heating and stirring, have a much lower melting point than the original HBA and HBD. As compared with conventional volatile organic solvents, DESs have many advantages, e.g., biodegradability, low toxicity [10], easy preparation [11], and other novel properties [12,13,14]. Because of these great properties, DESs have been extensively used to extract active ingredients from plants, such as the extraction of anthocyanins from *Catharanthus roseus* [15], hydrophilic and hydrophobic components from *Radix salvia miltiorrhizae* [16], bioactive flavonoids from *Equisetum palustre* L. [17], myricetin and amentoflavone from *Chamaecyparis obtuse* [18], artemisinin from *Artemisia annua* leaves [19], polysaccharides from *Dioscorea opposite* Thunb [20], and proanthocyanidins from *Ginkgo biloba* leaves and the evaluation of their antioxidant activity [21].

Baicalin, as the active ingredient of *Scutellaria baicalensis* Georgi, a traditional Chinese herbal medicine [22], has been demonstrated to have various pharmacological activities, i.e., decreasing blood pressure, anti-toxin, anti-fever, and reducing the risk of cardiovascular diseases [23,24,25,26]. Therefore, baicalin is widely used in medicine, health foods, functional products and cosmetics. Currently, baicalin is mainly extracted from *Scutellaria baicalensis* Georgi using traditional methods, such as hot reflux-assisted extraction (HRAE) and ultrasound-assisted extraction (UAE) [27,28,29,30]. These traditional extraction methods consume a great amount of organic solvents, take a long time, and have adverse effects on the environment. Li et al. recently reported the use of DES (ChCl-LA) for the extraction of baicalin from *Scutellaria baicalensis* Georgi by [31]. However, the extraction efficiency was relatively low. In our previous studies [32], a hydrophobic DESs (DecA: N_4444_-Cl)-based microwave-assisted extraction (MAE) was used for the efficient extraction of baicalin from *Scutellaria baicalensis Georgi*, with a maximum yield of 106.96 mg/g, quite close to that achieved by the Pharmacopoeia procedure (104.94 mg/g).

In order to further improve the extraction efficiency of baicalin, a highly efficient extraction method was developed by using DESs coupled with UPE. In this study, the most suitable DES for DESs-UPE was screened from among five different DESs. With the extraction yield as an index, a Box-Behnken design was employed to optimize the extraction parameters for DESs concentration, extraction pressure, liquid-solid ratio and extraction time. A comparative study with traditional extraction methods (e.g., HRAE, MAE) along with sample microstructure visualization using SEM were conducted to confirm the superiority of the DESs-UPE method.

## 2. Results and Discussion

### 2.1. Comparison of the Solubility

As an essential step of the extraction process, the dissolution of active ingredients from the herb was evaluated by equilibrium solubility. The compositions of the prepared DESs are listed in Table 1. 

As shown in Table 2, the equilibrium solubility of baicalin in different solvents at room temperature was determined for screening the best DES. The solubility of baicalin in water and 70% ethanol were used as the control. The equilibrium solubility of baicalin in DES-1, 22.7 mg/mL, was the greatest, which was approximately 200 times that in water and 20 times that in 70% ethanol. As expected, DESs have high solvency for water insoluble compounds [33]. In the DES matrix, the carboxylic acid, hydroxy group, and carbonyl group groups form a hydrogen-bonding network via intermolecular interactions. Baicalin is a flavone glycoside compound with several hydroxy groups, the high dissolution of DES-1 was attributed to strong intermolecular hydrogen-bonding between baicalin and DES-1.

Polarity is another important factor affecting solubilizing capacity [33]. As shown in Table 2, there is a large variation for the equilibrium solubility of baicalin in different DESs. The equilibrium solubility of baicalin in DES-1 was the best of all DESs. The bioactive compounds can be easily dissolved in the solvents having similar polarity. It is speculated that similar polarity between molecules of DES-1 and baicalin were responsible for their high dissolution.

### 2.2. Comparison of the Extractability of DESs

The extraction efficiency of five different DES was studied by UPE under the same extraction pressure (500 MPa), extraction time (5 min) and liquid-solid ratio (40 mL/g). From Figure 1, the DES extraction yield of baicalin varied with the different solvents. Using 70% ethanol, the most common solvent used for baicalin extraction as reference, DES-1 displayed clear superiority over other DESs with an extraction yield of 72 mg/g. The lower extraction yield of other DESs may be due to their high viscosity.

It is well known that the extraction efficiency of target compounds is affected by many factors, such as polarity, diffusion, solubility, viscosity, surface tension and physicochemical interaction with extraction solvents [34,35,36]. In other words, the type of DES plays a decisive role on the baicalin extraction efficiency. DES-1 had a better performance than other DESs, which may be due to a higher ability to form hydrogen bonds and more electrostatic interactions of DES-1 with baicalin than other DESs [37]. Another important factor for the extraction efficiency is the solvent polarity. Theoretically, the bioactive compounds can be easily extracted with solvents having similar polarity. DES-1 gave the highest extraction yield, and this suggested DES-1 has a similar polarity to baicalin [33]. The solvent viscosity has a great impact on the dissolution and diffusion of extracts. By visual observation, the viscosity of DES-1 was relatively less than other DESs. The lesser viscosity led to the relatively large diffusivity, and this resulted in a high extraction yield [31]. In summary, DESs as a mixture have complicated physical and chemical properties, when compared with protic organic solvents as pure substances. Therefore, it is difficult to find the rules for the extraction yield in DESs [38].

### 2.3. Optimization of the Extraction Conditions by RSM

From the above results, DES-1 was selected as the most appropriate DES for the process optimization to maximize the extraction yield of baicalin. In order to analyze the effect of various independent variables on the extraction rate, a single factor experiment was conducted to determine the main variables of Box-Behnken design (BBD). Four independent variables—water content (A), extraction time (B), extraction pressure (C) and liquid-solid ratio (D)—were determined and encoded at three levels. The associated negative signs (−1) was for low level, zero (0) represented the central value, and high level was expressed by a plus symbol (+1). The experimental design matrix and horizontal factors are shown in Table 3.

After the optimization, a BBD with 4-factors, 3-levels and 29 experimental runs was employed in this study. With the maximum extraction yield of baicalin as the response index, all response surface experiments and results are summarized in Table 4. 

The non-linear regression fitting based on the Quadratic model of the Design-Expert software (trial version 8.0.6.1, Stat-Ease Inc., Minneapolis, MN, USA) was employed to express baicalin extraction yield (Y_Baicalin_). The equation of dependent variables was shown below:Y_Baicalin_ = 115.95 − 0.4A + 2.6B + 2.51C − 1.76D − 1.05AB + 2.12AC + 1.1AD + 1.67BC − 5.24BD-3.22CD − 8.23A^2^ − 10.51B^2^ − 8.92C^2^ − 11.47D^2^(1)

The analysis of variance (ANOVA) for the quadratic models was listed in Table 5. The model was accurate and reliable by high coefficient determination (r^2^ = 0.9913) and non-significant lack-of-fit (*p* value of 0.1223). As shown in Table 5, the interaction terms BD and CD had a significant effect on the yield of baicalin, and the interaction (*p* < 0.0001) were significant. The “Pred R-Squared” of 0.976913 was in reasonable agreement with the “Adj R-Squared” of 0.991326.

Three-dimensional surface plots were used to study the interactions between the different variables of water content (A), extraction time (B), extraction pressure (C) and liquid to solid ratio (D) on the extraction yield of baicalin (Figure 2a–f). Generally, the relationship between the extraction yield and the single variable is described by a quadratic parabola. The results indicate that the extraction yield gradually increases with increasing water content, extraction time, extraction pressure and liquid to solid ratio, and then decrease after reaching the maximum. In summary, the extraction time, extraction pressure and liquid to solid ratio were more important than the water content. Figure 2a–c show that the water content has little effect on the extraction efficiency. As shown in Figure 2b, when the water content was fixed, the extraction yield gradually rose with increasing extraction pressure, reached a maximum at 400 MPa, and then slowly drop down. Therefore, the optimal pressure was considered as 400 MPa.

Higher pressure causes structural changes to the plant cells, and also increases the speed of solvent penetration into the plant material. This dual effect resulted in the intracellular active ingredient release from the disrupted cell wall. Therefore, the increase of extraction yield under increased pressure could be attributed to the acceleration of mass transfer. With pressure greater than 400 MPa, the cell wall and cell membrane were fully destroyed, and the target compounds completely dissolved.

Under this circumstance, the change of extraction yield was little. As shown in Figure 2d, the combined effect of extraction pressure and extraction time had more significant effect on the extraction yield. In Figure 2e, both the extraction time and liquid to solid ratio had more influence on the extraction yield. So the extraction yield gradually rose, and reached a maximum at 4 min and 110 mL/g. After reaching the maximum, the decrease of extraction yield was also slow. In Figure 2f, with the increase of extraction pressure and liquid to solid ratio, the extraction efficiency gradually reached the maximum, and then remained almost constant with slight drop-off.

The extraction conditions of baicalin were optimized by the regression analysis of model equation. The optimum conditions were determined as 39.22 vol% water content, extraction pressure of 418.3 MPa; extraction time of 4.15 min and a liquid-solid ratio of 109.23 mL/g. Under the optimum condition, the maximum extraction yield was 116.448 mg/g.

The validation test was conducted with the fixed optimum condition and the prediction value was determined by regression analysis. The final extraction parameters were determined as 40 vol% water content, extraction pressure of 400 MPa, extraction time of 4 min and a liquid-solid ratio of 110 mL/g, respectively. Under the fixed optimum conditions, parallel tests were repeated three times, and the average extraction yield obtained was 116.8 mg/g, which was consistent with the predicted value. These results confirmed that the response model could be used for the optimization2.4. Comparison of Extraction Methods

The comparison of extraction methods and extraction solvents were conducted and listed in Table 6. Two different extraction solvents in this study were 40% DESs-containing aqueous solutions and 70% ethanol, and three different methods were MAE, UPE and HRAE. The extraction efficiency from DES-based UPE as 116.8 mg/g is slightly higher than 110.4 mg/g from the Pharmacopoeia procedure. However, the extraction time is dramatically shortened from 3 h to 4 min. When compared with other reported baicalin extraction by DESs [31,32], the extraction efficiency from DES-based UPE is still better.

From the above results, the DESs-UPE method had a significant superiority over other methods on the extraction efficiency of baicalin. With the acoustic cavitation effect, the solvent penetration into plant cells was promoted, and the release of intracellular active ingredients from the disrupted cell walls was facilitated in the UPE process. As compared with traditional HRAE and MAE, the high pressure from UPE process increased the rate of dissolution and diffusion of baicalin from the plant cells. Furthermore, the DESs also could accelerate plant cell rupture for the release of the intracellular products [39,40,41,42]. Therefore, the current results revealed the DES-based UPE was a more rapid and efficient extraction method for the target compounds.

### 2.4. Microstructure Alteration of Different Extraction Procedures

In order to study the effect of different extraction methods on the microstructure of *Scutellaria baicalensis*, SEM was used to observe the raw and extracted *Scutellaria baicalensis* Georgi samples (Figure 3). As shown in Figure 3a, the external surface of the raw sample was smooth without apparent disruption on the cell surface. The hot reflux during HRAE extraction, only a few cells was slightly ruptured (Figure 3d,g) by the heat conduction and convection. Thus, the target compounds were extracted mainly through solubilization and permeation and the extraction time was long for HRAE. In the process of MAE, the sample was partially destroyed (Figure 3c,f). In the process of UPE, the pressure build-up within the plant cells probably exceeded their limit of contraction and caused them ruptured more rapidly. After UPE (Figure 3b,e), the surface of the cells was seriously destroyed with several holes on the walls, and some cells even broke into small fragments. This indicated that ultrahigh pressure resulted in a severe change in the surface of cells, which made the dissolution and diffusion more readily.

With the same extraction methods, the degree of disruption of plant cells followed the order: DESs-based HRAE (Figure 3d) > 70% ethanol-based HRAE (Figure 3g), DESs-based MAE (Figure 3c) > 70% ethanol-based MAE (Figure 3f), DESs-based UPE (Figure 3b) > 70% ethanol-based UPE (Figure 3e). These results indicated that the plant cells were easily disrupted in DESs conditions. The DESs caused damage may be attributed to the cell wall fiber dissolution in DESs [43]. Generally speaking, the degree of microstructure changes observed by SEM were consistent with the resultant extraction efficiency. The more ruptured cell resulted in a more efficient extraction. From the microscopic point of view, the severe fragmentation from DES-UPE rendered the dissolution and diffusion more easily, and then more efficient extraction.

## 3. Materials and Methods

### 3.1. Materials and Reagents

Dried *Scutellaria baicalensis* Georgi was purchased from the *Scutellaria baicalensis* Georgi Planting Base in Lingchuan County (Shanxi, China). It was ground, sifted using a 60-mesh stainless steel sifter, and then stored in sealed desiccators for use. Baicalin standard substance (>93.3%), was purchased from National Institutes for Food and Drug Control. Choline chloride (ChCl), glycerol (GL), ethylene glycol (EG), glucose (Glu), lactic acid (LA), 1,4-butanediol (BDO) and all other general reagents used were of analytical grade from Shanghai Yuanye Bio-Technology Co., Ltd. (Shanghai, China) and phosphoric acid and other organic solvents of HPLC grade were from MREDA Technology Inc. (Beijing, China). Water was deionized.

### 3.2. HPLC Analysis

The HPLC analysis protocol was developed in our previous publication [44], and there was almost no modification in this study. Quantitative HPLC analysis was performed on an Agilent 1200 chromatography system (Agilent Technologies, Waldbronn, Germany) equipped with a high-pressure gradient (G1311C), a VWD detector (G1314B), an auto-sampler (G1329B), a column oven (GT-30) and a Venusil XBP-C18 (4.6 mm × 250 mm, 5 µm, 100 Å) HPLC column. Chromatograms were recorded at 278 nm. The column temperature was maintained at 40 °C and the injection volume was 10 µL. The mobile phase is consisted of CH_3_OH (A) and 0.4% phosphoric acid (B) and the gradient elution was performed at a flow rate of 1 mL/min, 38% (A) for 0–50 min; 38–58% (A) for 50–55 min; 58% (A) for 50–70 min [44]. Chromatograms were scanned at 278 nm. All samples were filtered through 0.45 µm cellulose membranes prior to HPLC analysis. Acquisition and analysis of data were performed by Agilent Open LAB CDS Chemstation edition Software Ver. C. 01. 07 (Agilent Technologies).

### 3.3. Preparation of DESs

ChCl (HBA) was mixed with different types of HBD (GL, EG, BDO, LA and GLU) at a molar ratio of 1:1 in a sealed glass bottle. The mixture was then heated in a water bath at 80 °C until it became a transparent liquid and then cooled to room temperature for 12 h. DESs were further dried in a desiccator with P_2_O_5_ as desiccant. The prepared DESs are listed in Table 1. The water content in the DESs was 10, 20, 30, 40, 50 and 60 vol%, respectively. All the DESs solutions were homogeneous.

### 3.4. Sample Digestion and Analysis

Solubility tests were carried out using distilled water, 70% ethanol and DESs with an excess of the tested compound in a capped bottle and stirred at 40 °C for 2 h. The resulting mixture containing undissolved solid compounds were centrifuged. The supernatant was transferred to a 5 mL-microtube, diluted with methanol and vortexed to a homogeneous solution. The concentration of baicalin in the solution was monitored by HPLC. All solubility tests were performed in triplicate.

### 3.5. Extraction with Different Solvents

*Scutellaria baicalensis* Georgi was pulverized to finer than 40 mesh; a 0.25 g sample was then dispersed in the DESs in a two-neck pear shaped flask. The mixture was poured into a plastic bag, and the bag was sealed by heating and subjected to ultrahigh pressure of 500 MPa for 5 min. The extracted solution was collected and then filtered, and the supernatants were pooled and stored at 4 °C until HPLC analysis. All of the experiments were performed in triplicate.

### 3.6. Comparison Experiments

Different extraction processes were compared to evaluate the efficiency of the DES-based ultrahigh pressure extraction method. The following three common extraction techniques were selected as: UPE (400 MPa, 110 mL/g, 4 min), MAE (85 °C, 110 mL/g, 900 W for 4 min) and HRAE (80 °C, 133 mL/g, for 3 h, in the Chinese Pharmacopoeia)

## 4. Conclusions

In this study, the extraction of baicalin from *Scutellaria baicalensis* Georgi was investigated with a series of DESs prepared by mixing ChCl with different HBDs. DES with ChCl: LA at a 1:1 ratio has the best equilibrium solubility and extraction efficiency. From the optimization by RSM, the best extraction conditions were as follows: water content 40 vol%, extraction pressure 400 MPa, extraction time 4 min, and a liquid-solid ratio 110 mL/g, which gave an extraction yield of 116.8 mg/g. The developed DESs-based UPE method using ChCl: LA has been demonstrated to be more efficient than the traditional procedures (e.g., HRAE, MAE). Furthermore, the DESs-based UPE method is environmentally benign, less time-consuming, and highly-efficient. In conclusion, DESs-based UPE method has the great potential for an alternative extraction of baicalin from *Scutellaria baicalensis* Georgi, and may be applied for the extraction of bioactive compounds with similar structures from plant materials.

## Figures and Tables

**Figure 1 molecules-23-03233-f001:**
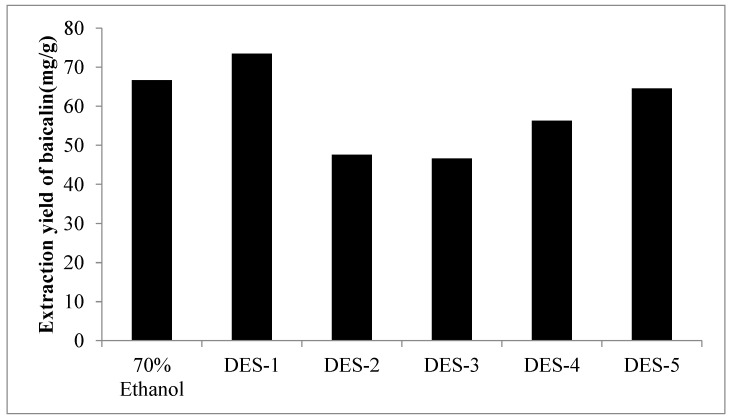
Extraction yield of baicalin using different types of DESs.

**Figure 2 molecules-23-03233-f002:**
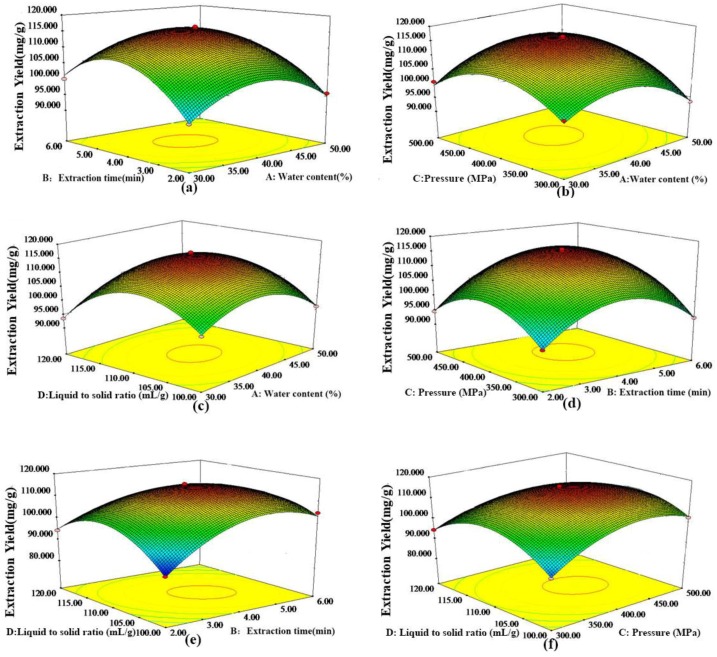
3D response surface plots of baicalin. (**a**) Varying extraction time and water content; (**b**) Varying extraction pressure and water content; (**c**) Varying liquid-solid ratio and water content; (**d**) Varying extraction pressure and time; (**e**) Varying liquid-solid ratio and extraction time; (**f**) Varying liquid-solid ratio and extraction pressure.

**Figure 3 molecules-23-03233-f003:**
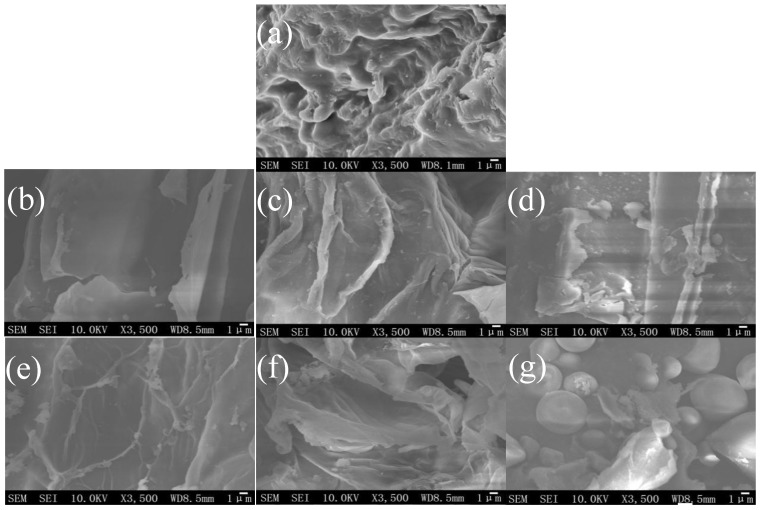
SEM images of *Scutellaria baicalensis* Georgi samples. (**a**) Raw materials; (**b**)DES-based UPE; (**c**) DES-based MAE; (**d**) DES-based HRAE; (**e**) 70% ethanol-based UPE; (**f**) 70% ethanol-based MAE; (**g**) 70% ethanol -based HRAE.

**Table 1 molecules-23-03233-t001:** Different composition of DESs applied in this work.

No.	Type of HBD	Abbreviation	ChCl/HBD Ratio
DES-1	Lactic acid	LA	1:1
DES-2	Glucose	GLU	1:1
DES-3	Glycerol	GL	1:1
DES-4	1,4-Butanediol	BDO	1:1
DES-5	Ethylene glycol	EG	1:1

**Table 2 molecules-23-03233-t002:** Solubility of baicalin in different solvents.

Number	Solvents	Solubility of Baicalin (mg/g)
1	Water	0.18
2	70% Ethanol	1.85
3	DES-1	22.7
4	DES-2	18.2
5	DES-3	12.35
6	DES-4	2.25
7	DES-5	17.8

**Table 3 molecules-23-03233-t003:** Variables in the Box-Behnken design.

Factor	Actual and Coded Levels Used for the Conditions		
	Low (−1)	Medium (0)	High (+1)
A = Water content (%)	30	40	50
B = Time (min)	2	4	6
C = Pressure	300	400	500
D = Liquid to solid ratio	100	110	120
Dependent variable	Constrains		
R1 = Baicalin/(mg/g)	Maximize		

**Table 4 molecules-23-03233-t004:** Box–Behnken design with independent variables and measured response.

Run	Factor A: Water Content %	Factor B: Time min	Factor C: Pressure MPa	Factor D: Liquid-Solid Ratio (mL/g)	Baicalin (mg/g)
1	40	2	400	100	88.08
2	40	2	300	110	93.08
3	50	6	400	110	98.03
4	40	6	300	110	94.79
5	40	2	500	110	94.52
6	40	4	300	120	94.58
7	30	4	300	110	99.60
8	40	4	400	110	116.560
9	50	4	400	100	96.35
10	50	2	400	110	95.989
11	40	4	500	120	93.04
12	30	4	400	120	93.56
13	40	4	400	110	116.05
14	30	2	400	110	93.84
15	30	4	400	100	99.05
16	40	4	400	110	115.88
17	50	4	500	110	103.09
18	40	4	400	110	115.98
19	40	6	400	120	90.22
20	30	6	400	110	100.06
21	40	6	500	110	102.91
22	50	4	300	110	93.43
23	50	4	400	120	95.26
24	40	4	300	100	91.18
25	40	6	400	100	104.95
26	30	4	500	110	100.78
27	40	4	400	110	115.26
28	40	4	500	100	102.51
29	40	2	400	120	94.31

**Table 5 molecules-23-03233-t005:** Box–Behnken design with independent variables and measured response.

Source	Sum of Squares	df	Mean Square	F Value	*p*-Value Prob > F	
Model	2022.273	14	144.4481	229.5803	<0.0001	significant
A-Water content	1.872498	1	1.872498	2.976078	0.1065	
B-Time	80.8396	1	80.8396	128.4834	<0.0001	
C-Pressure	75.90223	1	75.90223	120.6362	<0.0001	
D-Liquid-solid ratio	37.33898	1	37.33898	59.34518	< 0.0001	
AB	4.378975	1	4.378975	6.959778	0.0195	
AC	17.9536	1	17.9536	28.53477	0.0001	
AD	4.840466	1	4.840466	7.693255	0.0149	
BC	11.18712	1	11.18712	17.78039	0.0009	
BD	109.843	1	109.843	174.5803	<0.0001	
CD	41.42853	1	41.42853	65.84495	<0.0001	
A^2^	439.4122	1	439.4122	698.3853	<0.0001	
B^2^	716.8069	1	716.8069	1139.266	<0.0001	
C^2^	516.0715	1	516.0715	820.2247	<0.0001	
D^2^	853.7272	1	853.7272	1356.882	<0.0001	
Residual	8.808564	14	0.629183			
Lack of Fit	7.89223	10	0.789223	3.445136	0.1223	not significant
Pure Error	0.916333	4	0.229083			
Cor Total	2031.081	28				
R-Squared			0.995663			
Adj R-Squared			0.991326			
Pred R-Squared			0.976913			

**Table 6 molecules-23-03233-t006:** Extraction contents of the baicalin using various extraction procedures.

Extract Method	Solvent	Baicalin (mg/g)
UPE	70% Ethyl alcohol	108.6
	Moisture content 40% of ChCl-LA (1:1)	116.8
HRAE	70% Ethyl alcohol	110.4
	Moisture content 40% of ChCl-LA (1:1)	84.3
MAE	70% Ethyl alcohol	89.3
	Moisture content 40% of ChCl-LA (1:1)	101.5
MAE	Moisture content 20% of ChCl-LA (1:2)	33.1 [31]
MAE	Moisture content 33% of DecA-N_4444_-Cl (1:2)	106.96 [32]

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
