# Peer review of "Deep Eutectic Solvent-Based Ultrahigh Pressure Extraction of Baicalin from Scutellaria baicalensis Georgi"

_molecules, 2018, doi:10.3390/molecules23123233_

Reviewer 1 Report

There are lots of spelling mistakes in the manuscript, as well as capital letters in the middle of the sentence, for example: "Because of sucn good properties, there are ranges of reports on the use of DESs to extract active ingredients from plants, Such as extraction of anthocyanins from Catharanthus roseus [13], Hydrophilic and Hydrophobic Components from Radix Salviae miltiorrhizae [14],Bioactive flavonoids from Equisetum palustre L[15]." Those terms were simply copy-pasted from the titles of the manuscripts. this manuscript is full of this kind of mistakes, which should be corrected. Also, constantly they write enthnol instead of ethanol. 

Is DES-1 combination of choline chloride and lactate or lactic acid? 

The authors claim that the solubility of the target compound is affected by physical properties of DESs, but it should be explained how. Do they have the data for viscosity and surface tension of given DESs? If so, it should be stated or at least compared to the literature data.

In Figure 1, an amount of baicalin can not be stated as extraction rate.

In section starting with line 98, the authors discuss about polarity and viscosity of DESs, but not a single reference of actual polarity or viscosity is referred to or viscosity or polarity determined.   

Molar ratio of HBA and HBD is in all cases 1:1. In the literature some ratios are often 1:2, like for ChCl:EG, ChCl:BDO, so it would be appreciated if author can explain why they chose this ratio.

There are no validation parameters for HPLC analysis and those should be added to the Section 3.2.

Overall impression is that authors could have extended their research to more deep eutectic solvents which would have given a better insight into their extraction potential. They have limited their investigation on only one carboxylic acid and one sugar as HBD and other HBDs are alcohols. If an actual screening is to be performed, at least one of each HBD groups should be included (for example there are no amides, like urea and thiourea taken into consideration). These are the major drawbacks of this manuscript.

Some good points of the manuscript are an extensive optimization which was performed, as well as comparison of different extraction methods and determination of microstructure, which gives some added value to the manuscript.

So, if authors can rewrite their manuscript according to the comments and extensively check it regarding spelling and grammar it can be accepted for publication.

Author Response

Response to Reviewer 1 Comments

Point 1: There are lots of spelling mistakes in the manuscript, as well as capital letters in the middle of the sentence, for example: "Because of sucn good properties, there are ranges of reports on the use of DESs to extract active ingredients from plants, Such as extraction of anthocyanins from Catharanthus roseus [13], Hydrophilic and Hydrophobic Components from Radix Salviae miltiorrhizae [14], Bioactive flavonoids from Equisetum palustre L[15]." Those terms were simply copy-pasted from the titles of the manuscripts. this manuscript is full of this kind of mistakes, which should be corrected. Also, constantly they write enthnol instead of ethanol.

Response 1: Thanks to the reviewer for this suggestion. The entire manuscript has been double-proofread by a colleague with advanced education from American university. We have taken this seriously and asked a colleague with PhD from US university to proofread the entire manuscript. The majority of changes have been marked in the manuscript as red.

Point 2: Is DES-1 combination of choline chloride and lactate or lactic acid? 

Response 2: Thanks to the reviewer for this suggestion. This is a careless mistake. DES-1 is composed of choline chloride and lactic acid, which has been corrected in the revised paper.

Point 3: The authors claim that the solubility of the target compound is affected by physical properties of DESs, but it should be explained how. Do they have the data for viscosity and surface tension of given DESs? If so, it should be stated or at least compared to the literature data.

Response 3: We are very grateful for the reviewer’s comments. The original explanations may have some irrationality. Two factors may have important impact on the solubility of baicalin in DESs. The first one is the ability of forming hydrogen bonding between baicalin and DES. The second one is “like dissolves like” that means polar solutions dissolve polar substances and non-polar solutions dissolve non-polar substances. The viscosity and surface tension may have less effect on the solubility of baicalin. Therefore, this part of the manuscript has been heavily edited and marked as red in the section of “2.1. Comparison of the Solubility”.

Point 4: In Figure 1, an amount of baicalin can not be stated as extraction rate.

Response 4: Thanks to the reviewer for this suggestion. This expression “extraction rate” is due to the language barrier. All “extraction rate” have been changed into “Extraction yield” in the revised paper.

Point 5: In section starting with line 98, the authors discuss about polarity and viscosity of DESs, but not a single reference of actual polarity or viscosity is referred to or viscosity or polarity determined.  Response 5: Thanks to the reviewer for this suggestion. This part has been heavily edited and included 6 references in the revised manuscript marked as red.

Point 6: Molar ratio of HBA and HBD is in all cases 1:1. In the literature some ratios are often 1:2, like for ChCl:EG, ChCl:BDO, so it would be appreciated if author can explain why they chose this ratio.

Response 6: Thanks to the reviewer for this suggestion. In our preliminary experiment, our results with ChCl-LA(a ratio of 1:1)is very close to the extraction yield of pharmacopoeia procedure, and after the optimization of other factors, the extraction yield has reached the ideal level. Therefore, the ratio of 1:1 was for this study. It is a good suggestion to select other molar ratio of HBA and HBD, I believe that we can achieve better results. We will do this in our future work. Thanks for your understanding.

Point 7: There are no validation parameters for HPLC analysis and those should be added to the Section 3.2.

Response 7: Thanks to the reviewer for this suggestion. The development of HPLC analysis for baicalin and the validation have been reported in our previous publication (Journal of Chromatography B, 2015, 1002, 411–417). In this study, the major focus was to find the more efficient protocol for the extraction of baicalin, and the HPLC analysis was the almost same as the previous report. This has been clarified in the revised manuscript and marked as red.

Point 8: Overall impression is that authors could have extended their research to more deep eutectic solvents which would have given a better insight into their extraction potential. They have limited their investigation on only one carboxylic acid and one sugar as HBD and other HBDs are alcohols. If an actual screening is to be performed, at least one of each HBD groups should be included (for example there are no amides, like urea and thiourea taken into consideration). These are the major drawbacks of this manuscript.

Response 8:  Thanks to the reviewer for this suggestion. The investigation of DESs was limited to carboxylic acid, sugar and alcohols as HBD, and no amides, like urea and thiourea were taken into consideration. The main consideration is that the lactic acid-based DES-UPE extraction of flavonoid glycosides has reached the expected efficiency when compared with the extraction efficiency by conventional solvents. It is interesting that we are of exactly the same idea as the reviewer. Actually, at first, we have tried to select urea/thiourea as HBD. However, the urea/thiourea can be irritating to skin, eyes, and the respiratory tract. Repeated or prolonged exposure to urea/thiourea may cause dermatitis, and respiratory discomfort. In case this protocol is used for the large amount extraction of flavonoid glycosides, it is inevitable to have a prolonged exposure to urea/thiourea. So, the screening of urea/thiourea as HBD had to give up. Thanks for your understanding.

Reviewer 2 Report

The present ms. from Zhang and co-workers investigates the extraction of baicalin from Scutellaria Baicalensis Georgi by synergystically combining the ultrahigh pressure extraction technique with a ChCl/lactate eutectic mixture, as an environmentally friendly reaction medium. The influence of DES concentration, extraction pressure, extraction time and the solid/liquid ratio have been carefully evaluated so as to isolate, under optimized conditions, baicalin in a yield to up to 116.8 mg/g.  

The work has been competently researched, and this reviewer considers the story and data quality of this ms. would be of an acceptable standard for publication in Molecules after the following revision.

(a) There are more recent reviews on the usefulness of DESs in the extraction field that can be included.

(b) It would be helpful for the reader to include in Table 6 also the results obtained by Li (ref. 27) and by the same authors (ref. 28) when using hydrophobic DESs, and compare the efficiency and pros/cons of the various methodologies in terms of extraction yield.

(c) English needs brushing up throughout the ms. There are several flaws. Consider, for example, the following: “It has decreasing blood pressure (??)…” “Enthnol” “Because of sucn good properties…”; there is several times a capital letter after a comma, etc.

Author Response

Response to Reviewer 2 Comments

Point 1:  There are more recent reviews on the usefulness of DESs in the extraction field that can be included.

Response 1: Thanks for this suggestion. Some recent literatures [8, 9, 20, 21] have been added in the revised manuscript.

Point 2: It would be helpful for the reader to include in Table 6 also the results obtained by Li (ref. 27) and by the same authors (ref. 28) when using hydrophobic DESs, and compare the efficiency and pros/cons of the various methodologies in terms of extraction yield.

Response 2: Thanks to the reviewer for this suggestion. The results obtained by Li (ref. 27, renumbered as 31 in the revised edition) and by our group (ref. 28, renumbered as 32) have been included in Table 6. The relative description about the efficiency and pros/cons of the various method was added in the revised manuscript marked as red.

Point 3: English needs brushing up throughout the ms. There are several flaws. Consider, for example, the following: “It has decreasing blood pressure (??)…” “Enthnol” “Because of suc 

Response 3: Thanks to the reviewer for this suggestion. We have taken this seriously and asked a colleague with PhD from US university to proofread the entire manuscript. The majority of changes have been marked in the manuscript as red.

Round  2

Reviewer 1 Report

The authors complied with all the suggestions, but the manuscript still needs some improvements considering English language. After that it will be suitable for publication.